# Fatty Liver and Risk of Head and Neck Cancer in Type 2 Diabetes Mellitus: A Nationwide Cohort Study

**DOI:** 10.3390/cancers15041209

**Published:** 2023-02-14

**Authors:** Junhee Park, Kyungdo Han, Seung Woo Lee, Yeong Jeong Jeon, Sang-Man Jin, Wonyoung Jung, Yoon Kyoung So, Sang Duk Hong, Dong Wook Shin

**Affiliations:** 1Department of Family Medicine/Supportive Care Center, Samsung Medical Center, Sungkyunkwan University School of Medicine, Seoul 06351, Republic of Korea; 2Department of Statistics and Actuarial Science, Soongsil Universiy, Seoul 06978, Republic of Korea; 3Department of Biostatistics, College of Medicine, Catholic University of Korea, Seoul 06591, Republic of Korea; 4Department of Thoracic and Cardiovascular Surgery, Samsung Medical Center, Sungkyunkwan University School of Medicine, Seoul 06351, Republic of Korea; 5Department of Endocrinology and Metabolism, Samsung Medical Center, Sugnkyunkwan University School of Medicine, Seoul 06351, Republic of Korea; 6Department of Otorhinolaryngology-Head & Neck Surgery, Ilsan Paik Hospital, Inje University College of Medicine, Goyang-Si 10380, Republic of Korea; 7Department of Otolaryngology, Head and Neck Surgery, Samsung Medical Center, Sungkyunkwan University School of Medicine, Seoul 06351, Republic of Korea; 8Department of Clinical Research Design and Evaluation, Samsung Advanced Institute for Health Science and Technology, Sungkyunkwan University, Seoul 06351, Republic of Korea

**Keywords:** diabetes mellitus, head and neck cancer, squamous cell carcinoma, non-alcoholic fatty liver disease, fatty liver index

## Abstract

**Simple Summary:**

The incidence of non-alcoholic fatty liver disease (NAFLD) is >2-fold higher in patients with diabetes mellitus (DM). NAFLD is thought to increase the risk of extrahepatic cancer; however, existing data on NAFLD and the risk of head and neck cancer (HNC) are limited. This is the first study to report the associations between NAFLD and HNC in Type 2 DM (T2DM) patients. Using a population-based retrospective cohort study with a total of 1,904,376 subjects with T2DM who received a health checkup during 2009–2012, NAFLD was associated with an increased risk of developing HNC in the oral cavity, pharynx, and larynx, but not in the salivary gland. Future investigations which would determine the underlying mechanism and how improvement of NAFLD can reduce the development of HNC are necessary.

**Abstract:**

This study is aimed at investigating the association between NAFLD and the risk of HNC separately based on cancer site using a large population-based cohort of patients with T2DM. The data used in this population-based retrospective cohort study were provided by the Korean National Health Insurance Service. The Cox proportional hazards model was used to estimate multivariable adjusted hazard ratios and 95% CIs for the association of the fatty liver index (FLI) and the risk of HNC. During the mean 6.9 years of follow-up, approximately 25.4% of the study cohort had NAFLD, defined as an FLI ≥60. A total of 3543 HNC cases were identified. Overall, patients with a higher FLI had a significantly higher risk of HNC in the oral cavity, pharynx, and larynx compared with patients with an FLI <30. An association was not observed between salivary gland cancer and FLI. There was no association between obesity and HNC. However, obese patients showed a lower risk of cancer for the oral cavity (*p* = 0.040), pharynx (*p* = 0.009), and larynx (*p* < 0.001) than non-obese patients with the same FLI level. Neither obesity nor smoking affected the association between FLI- and HNC-risk in stratified analyses. In T2DM patients, NAFLD was associated with an increased risk of developing HNC in the oral cavity, pharynx, and larynx, but not in the salivary gland.

## 1. Introduction

Approximately 900,000 cases of head and neck cancer (HNC) worldwide and 5000 cases of HNC in Korea are newly diagnosed each year [1]. In addition, HNC accounts for 5% of cancer mortality globally [1]. Identifying modifiable cancer risk factors is important because global cancer incidence and cancer death are rapidly increasing [1]. In several previous studies, the association between diabetes mellitus (DM) and HNC has been reported [2], although some showed weak [3] or no association [4]. A recent meta-analysis showed type 2 DM (T2DM) patients had an approximately 15% increased risk of cancer development, such as oral cavity and oropharyngeal cancer (SRR = 1.15, 95% confidence interval, CI 1.02–1.29; P_heterogeneity_ = 0.277, I^2^ = 15.4%; 10 studies) [5]. In a meta-analysis of four case-control studies, a positive association between T2DM and risk of oral precancerous lesion (SRR = 1.85, 95% CI 1.23–2.80; P_heterogeneity_ = 0.038, I^2^ = 57.5%) was also reported [5]. The global prevalence of non-alcoholic fatty liver disease (NAFLD) in DM is 55.5%, ≥2-fold higher than in the general population [6]. With the increasing prevalence of NAFLD in parallel with diabetes [7,8], the possible association between NAFLD and cancer development has been suggested in several previous studies [9,10,11]. In addition, squamous cell carcinoma as well as adenocarcinoma were shown to be associated with NAFLD [12]. Although NAFLD has less frequently been shown to be associated with HNC compared with other extrahepatic malignancies, the relationship with HNC (adjusted hazard ratio, aHR 1.36, 95% CI 0.85–2.17) has been reported in only one study [13]. However, the risk based on cancer site (lip, oral cavity, and pharynx) was not separately assessed and the association did not appear significant due to the small number of cancer cases (*n* = 40) [13]. Therefore, in the present study, the association between NAFLD and the risk of HNC was separately investigated based on cancer site using a large population-based cohort of patients with T2DM.

## 2. Materials and Methods

### 2.1. Study Design, Setting, and Population

The South Korean government manages the public medical insurance system. The Korean National Health Insurance Service (NHIS) is a mandatory universal insurance system that covers almost the entire population (97%) of South Korea. The other 3% of the population in the lowest income bracket are covered by government aid, such as Medicaid beneficiaries. The NHIS database contains data regarding the demographic factors of enrollees, diagnosis statements based on International Classification of Disease, 10th revision (ICD-10) codes and prescriptions [14]. The NHIS also provides a biennial national health check-up program for all beneficiaries >40 years of age and all employees regardless of age. The health check-up program includes a self-administered health questionnaire, including lifestyle behaviors (e.g., smoking, alcohol consumption, and exercise), as well as anthropometric measurements and laboratory tests (e.g., serum glucose, lipid profile, and liver enzymes) [15]. Patients with T2DM >20 years of age and who underwent health examinations during 2009–2012 were included in this study (*n* = 2,745,688). T2DM was defined as at least one claim per year based on the ICD-10 codes E11–14, with a prescription history of antidiabetic medication or a fasting glucose level ≥126 mg/dL at the health screening examination. Type 1 DM was not included. Subjects diagnosed with liver cirrhosis (*n* = 59,842), hepatitis (*n* = 408,660), any cancer (*n* = 57,636), or possessing any missing data (*n* = 75,544) were excluded. Heavy alcohol drinkers consuming ≥30 g/day for males and ≥20 g/day for females (*n* = 224,409) were also excluded according to the Korean Association for the Study of the Liver Clinical Practice Guideline for NAFLD [16]. A 1-year lag time was applied to reduce the effect of reverse causality (*n* = 15,221 excluded). This resulted in 1,904,376 individuals for final analysis. The Institutional Review Board (IRB) of Soongsil University approved this study (IRB File No. SSU-202003-HR-201-01).

### 2.2. Definition of NAFLD

NAFLD was assessed based on the fatty liver index (FLI), a non-invasive surrogate predictor calculated using the following formula based on waist circumference (WC), body mass index (BMI), triglyceride (TG), and gamma-glutamyl-transferase (GGT):$$\mathrm{FLI}=\frac{e^{0.953\times \ln\left(\mathrm{TG}\right)+0.139\times \;\mathrm{BMI}\;+0.718\times \ln\left(\mathrm{GGT}\right)+0.053\times \;\mathrm{WC}\;-15.745}}{1+e^{0.953\times \ln\left(\mathrm{TG}\right)+0.139\times \;\mathrm{BMI}\;+0.718\times \ln\left(\mathrm{GGT}\right)+0.053\times \;\mathrm{WC}\;-15.745}}\times 100$$

An FLI ≥60 was previously shown to be an accurate diagnostic marker for NAFLD and practical for screening the general population in epidemiological studies. FLI scores range from 0–100 and are divided into three levels: <30, 30–<60, and ≥60 [17].

### 2.3. Study Outcomes

Newly developed HNC was defined operationally based on the diagnosis code and registration specific to the national special co-payment program. Specific HNC sites were classified as the oral cavity (C00–C06), salivary gland (C07, C08), pharynx (C09–C13), and larynx (C32). In Korea, cancer patients can apply for a special co-payment reduction program with a medical certificate of cancer diagnosis from their physician which allows them to pay only 5% of medical expenses for cancer work-up and treatment. Because nearly all cancer patients are registered in this program in Korea, the data on HNC diagnosis used in the present study were sufficiently reliable [18].

### 2.4. Covariates

BMI was calculated using weight (kg) divided by height in meters squared (m^2^), and individuals were classified into one of five groups: underweight (<18.5 kg/m^2^), normal (18.5–22.9 kg/m^2^), overweight (23–24.9 kg/m^2^), obese I (25–29.9 kg/m^2^), and obese II (≥30 kg/m^2^), according to Asia-Pacific guidelines from the Western Pacific Regional Office (WPRO). Comorbidities were defined using the medical claims data before screening based on ICD-10 codes and relevant medication (for hypertension, I10–I13 or I15 and antihypertensive drugs or blood pressure ≥140/90 mmHg; for dyslipidemia, E78 and lipid-lowering drugs or total cholesterol level ≥240 mg/dL). Cigarette smoking, alcohol consumption, and regular exercise behavior data were also collected. Smoking was classified based on smoking status (non-, ex-, and current smokers) and smoking amount in pack-years (PYs), according to a previous study [19]. The daily amount of alcohol intake was calculated by multiplying the average frequency of alcohol intake (per week) and the typical number of standard drinks on each occasion. Alcohol consumption was classified into none and mild. Regular exercise was defined as >30 min of moderate physical activity ≥5 times per week or >20 min of strenuous physical activity ≥3 times per week.

### 2.5. Statistical Analysis

Continuous variables are presented as mean ± standard deviation (SD) and categorical variables as number and percentage. The ANOVA was used to determine differences between the means of continuous variables and the chi-square test was used to evaluate differences in the proportion of categorical variables. HR and 95% CI values for HNC incidence were analyzed using the Cox proportional hazards model. The proportional hazards assumption was evaluated using Schoenfeld residuals. A multivariable model was adjusted for age, sex, BMI, alcohol consumption, regular physical activity, household income, and DM-related information such as disease duration, glucose level, insulin use, and the number of oral antihyperglycemic (OHA) drugs. Stratified analyses based on obesity and smoking status were performed to determine potential interaction between FLI and these variables on the incidence of HNC. Statistical analyses were performed using SAS version 9.4 (SAS Institute Inc., Cary, NC, USA), and a *p*-value < 0.05 was considered statistically significant.

## 3. Results

### 3.1. Characteristics of Study Participants

Table 1 shows the characteristics of the study participants categorized into three groups based on the FLI. Among the 1,904,376 T2DM patients, approximately 25.4% (*n* = 484,300) had NAFLD defined as an FLI ≥60. The mean age in this group was 54.1 ± 12.2 years and 71.5% were male. When compared to the FLI <30 group, the subjects in the FLI ≥60 group were more likely to be a current smoker, alcohol consumer, physically inactive, obese, and possessing comorbidities with hypertension or dyslipidemia. The subjects in the FLI ≥60 group also displayed higher weight, WC, BMI, blood pressure, glucose, total cholesterol, and TG. The individuals in the FLI ≥60 group had a shorter duration (<5 years) of DM and no treatment of insulin or OHA with <3 drugs. Study participants who developed HNC were older and more likely to be a current smoker and alcohol consumer (Appendix A).

### 3.2. Association between FLI and HNC

A total of 3543 HNC cases were identified during the mean 6.9 years of follow-up: oral cavity (*n* = 949), pharynx (*n* = 1152), larynx (*n* = 1085), and salivary gland (*n* = 357). Table 2 shows the association between FLI and HNC in DM patients. Overall, subjects with a higher FLI had a significantly higher risk of HNC in the oral cavity (aHR 1.24, 95% CI 1.05–1.46 in FLI 30–59; aHR 1.33, 95% CI 1.06–1.66 in FLI ≥60), pharynx (aHR 1.30, 95% CI 1.12–1.51 in FLI 30–59; aHR 1.46, 95% CI 1.20–1.79 in FLI ≥60), and larynx (aHR 1.11, 95% CI 0.95–1.29 in FLI 30–59; aHR 1.42, 95% CI 1.16–1.74 in FLI ≥60) compared with subjects with an FLI <30. Association was not observed between salivary gland cancer and FLI.

### 3.3. HNC Risk Based on BMI and FLI

Table 3 shows the association between the risk of HNC and the combination of obesity and FLI. The risk of oral cavity (*p* = 0.040), pharynx (*p* = 0.009), and larynx (*p* < 0.001) cancer was greater in non-obese individuals in the higher FLI groups than in non-obese individuals in the FLI <30 group when used as a reference. Obese patients showed a lower risk of cancer than non-obese patients in the FLI <30, with a decreased risk of pharynx (aHR 0.78, 95% CI 0.57–1.08 in FLI <30; aHR 0.91, 95% CI 0.76–1.08 in FLI 30–59; aHR 0.94, 95% CI 0.80–1.12 in FLI ≥60) and larynx cancer (aHR 0.80, 95% CI 0.57–1.12 in FLI <30; aHR 0.72, 95% CI 0.59–0.87 in FLI 30–59; aHR 0.91, 95% CI 0.77–1.08 in FLI ≥60). An association was not observed for salivary gland cancer.

### 3.4. HNC Risk Based on Age and FLI

Table 4 shows the association between the risk of HNC and the combination of age and FLI. There was no significant interaction across the age groups.

### 3.5. Stratified Analysis Based on Smoking Status and BMI

When the data were analyzed with a stratification based on obesity (≥25 kg/m^2^), no significant difference was found in the relationship between FLI and risk of all HNCs in obese and non-obese individuals. No significant difference was observed between FLI and the risk of all HNCs in currently smoking and not-currently smoking individuals. Neither obesity nor smoking status affected the association between FLI and HNC risk based on the stratified analyses (Figure 1).

## 4. Discussion

Although tobacco, alcohol, and viral exposure are known risk factors for HNC, minimal data exists on NAFLD and HNC risk [20]. To the best of our knowledge, this is the first study in which the association of NAFLD with the risk of HNC in patients with T2DM has been reported. This was a large, population-based, representative sample study. A significant number of HNC cases (*n* = 3543) was included, allowing detailed analysis based on NAFLD and cancer type.

### 4.1. Oral Cavity, Pharynx, and Larynx Cancer

Most HNCs are derived from the mucosal epithelium in the oral cavity, pharynx, and larynx, and are collectively known as head and neck squamous cell carcinoma (HNSCC) [21]. In the present study, a positive association was found between NAFLD and HNSCC with a dose-response pattern in T2DM patients. Although how patients with NAFLD develop cancers remains unclear, we hypothesize several putative mechanisms. The mammalian target of the rapamycin (mTOR) pathway is coupled to the insulin/IGF-1 signaling pathway; thus, NAFLD has a high probability of occurring due to hyperinsulinemia [22]. Insulin activates the insulin receptor (IR), which triggers the IR substrate (IRS) and phosphatidylinositol 3-kinase (PI3K), resulting in the phosphorylation of mTOR, eventually activating AKT [22,23]. The PI3K/AKT/mTOR pathway has a significant role in tumorigenesis in oral squamous cell carcinoma [24]. Targeted therapy based on cell surface signaling receptors currently includes wortmannin (LY294002), perifosine, and rapamycin [25]. Adenosine monophosphate-activated protein kinase (AMPK) is a ubiquitous energy-sensing enzyme within cells that is critical for maintaining metabolic homeostasis [26]. The inhibitory actions of activated AMPK on the downstream effects of mTOR signal transduction have been well described in previous studies [27]. Adipose tissue damage leads to AMPK inhibition, subsequently promoting NAFLD [26,27]. Thus, NAFLD is a condition in which adipose tissue damage leads to the activation of the mTOR signaling pathway. Furthermore, NAFLD is characterized by the excess accumulation of TGs in the hepatocyte due to both the influx of free fatty acids and de novo lipogenesis [28]. Fatty acid synthase (FAS) has a major role in the hepatic de novo lipogenesis and is an attractive target for NAFLD treatment [29]. Increased FAS gene expression in adipose tissue is associated with visceral fat accumulation, impaired insulin sensitivity, as well as increased circulating fasting insulin, interleukin-6 (IL-6), and leptin levels, indicating the important role of lipogenic pathways [30]. Notably, results of a previous study indicated that FAS is required for the proliferation of human oral squamous carcinoma cells [31]. Orlistat, an irreversible inhibitor of FAS that functions as a central regulator of lipid metabolism, reduces the growth and metastasis of oral tongue squamous cell carcinoma [32].

### 4.2. Obesity vs. Central Obesity

In the present study, obesity, defined by BMI and HNC (oral cavity, pharynx, and larynx cancer), showed no association (Appendix A). Currently, the association between obesity and the risk of HNC remains controversial [33,34,35]. Ward et al. noted that central adiposity, but not BMI, was associated with HNC [36]. In 2018, the World Cancer Research Fund (WCRF) concluded that higher body fat showed significant positive association with HNC for never-smokers from a pooled analysis of 20 cohort studies [37]. NAFLD appears to be a better predictor than BMI in reflecting an obesity phenotype with a higher malignancy potential because it is closely associated with central adiposity and insulin resistance [38]. In addition, among fatty liver index components, GGT was the most prominent biomarker of liver fat (Appendix A). This is because GGT, an enzyme responsible for the extracellular catabolism of antioxidant glutathione, may be linked to greater oxidative stress, which has been implicated in insulin resistance and diabetes [39]. Further research to determine the relationship between NAFLD and HNC will aid in understanding the detailed mechanism of insulin resistance and lipogenesis in central adiposity.

### 4.3. Stratified Analyses Based on Smoking Status and BMI

In the present study, neither obesity nor smoking affected the association between FLI and HNC risk based on stratified analysis. NAFLD and contributing factors (obesity and smoking) showed no synergistic or antagonistic effect on HNC. Therefore, we propose that NAFLD is an independent risk factor for HNC.

### 4.4. Salivary Gland Cancer

Conversely, salivary gland cancer was not associated with NAFLD in the present study. This is the first analysis in which the incidence of salivary gland cancer in relation to NAFLD has been reported. Salivary gland cancer includes heterogeneous tumors of different histological subtypes such as adenoid cystic carcinoma (ACC), mucoepidermoid carcinoma (MEC), mammary analog secretory carcinoma (MASC), carcinoma ex pleomorphic adenoma (Ca-ex-PA), and acinic cell carcinoma (AciCC) [40]. The annual incidence of salivary gland cancer is estimated at approximately 3 cases per 100,000 and accounts for 6–8% of all HNC. According to previous studies, viral infections (e.g., EBV, HIV), cervicofacial radiotherapy, occupational exposure to radioactive materials such as nickel compounds, and rubber manufacturing, were strong risk factors for salivary gland cancer [40]. However, most investigations of salivary gland cancer on exposures that are risk factors of HNSCC, such as obesity, smoking, and alcohol, have shown no significant association [41], which is consistent with our findings. Different histological types among HNCs might have resulted in different relationships to various exposures. Due to the rarity and variable pathologies of salivary gland cancer, knowledge regarding risk factors is limited. Future research with a larger number of cases among the subtypes of salivary gland cancer is necessary to elucidate the relationship.

### 4.5. Limitations

The present study had several limitations that should be considered. First, the findings might be subject to selection bias. Because the study was limited to subjects who had health screening examinations, the participants might tend to have healthier lifestyles (e.g., lower smoking rate) than the general population and, therefore, lower HNC risk. Second, information regarding NAFLD could not be collected from ultrasonography or liver biopsy, the gold standards for diagnosis of fatty liver. However, FLI is an effective surrogate and a highly predictive method which can be performed in routine medical visits [42]. FLI is widely considered an acceptable alternative to imaging modalities in clinical practice guidelines [42]. Third, the possibility of residual confounding variables, such as viral exposure, cannot be excluded. Whether the HNC was associated with HPV could not be determined from the data. Fourth, we could not compare the incidence of HNC between Type 1 DM and T2DM patients, as we only included T2DM. It is possible that the two different types of DM may have different associations with HNC risk, given the different pathophysiological mechanism between them. Finally, the study population was Korean and further studies are required to confirm the relationship between NAFLD and HNC for other ethnic backgrounds. Despite these limitations, this is the first study in which the associations between NAFLD and HNC in T2DM patients has been reported.

## 5. Conclusions

In conclusion, NAFLD has been shown to be associated with an increased risk of developing oral cavity, pharynx, and larynx cancer of HNC, but not of salivary gland cancer in DM patients. Future investigations are necessary to determine the underlying mechanism and how improvement of NAFLD can reduce the development of HNC in T2DM patients.

## Figures and Tables

**Figure 1 cancers-15-01209-f001:**
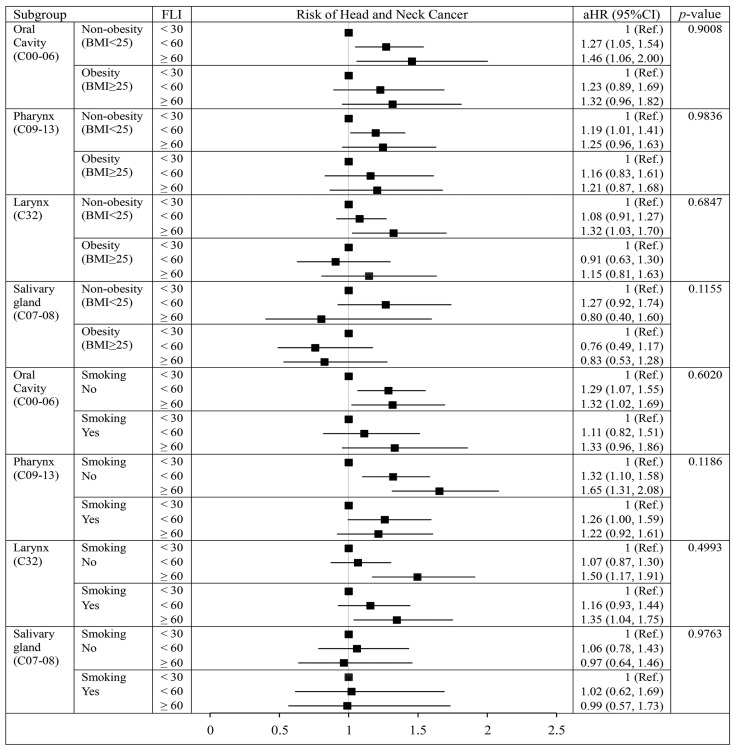
Forest plot of adjusted hazard ratios of head and neck cancer according to obesity and current smoking. BMI, body mass index; aHR, adjusted hazard ratio; CI, confidence interval. Values are adjusted for age, sex, BMI, smoking, alcohol consumption, regular physical activity, income, glucose, insulin, oral hypoglycemic agent, and diabetes mellitus duration.

**Table 1 cancers-15-01209-t001:** General characteristics of the subjects based on the risk of fatty liver among DM patients.

Variables	FLI			*p*-Value
	<30	30–59	≥60	
Total, *n* (%)	775,013 (40.7)	645,063 (33.9)	484,300 (25.4)	
Age, years, mean ± SD	58.9 ± 13.0	58.6 ± 11.8	54.1 ± 12.2	<0.0001
Age, years				<0.0001
<30	16,255 (2.1)	5120 (0.8)	7414 (1.5)	
30–<40	40,647 (5.2)	30,695 (4.8)	49,939 (10.3)	
40–<50	116,822 (15.1)	104 453 (16.2)	115,940 (23.9)	
50–<60	202,852 (26.2)	187,066 (29.0)	146,194 (30.2)	
60–<70	216,011 (27.9)	186,714 (29.0)	105,589 (21.8)	
70–<80	149,922 (19.3)	113,800 (17.6)	52,856 (10.9)	
≥80	32,504 (4.2)	17,215 (2.7)	6368 (1.3)	
Sex (male)	349,719 (45.1)	376,757 (58.4)	346,185 (71.5)	<0.0001
Income (lowest quartile)	186,563 (24.1)	149,428 (23.2)	111,981 (23.1)	<0.0001
Smoking status				<0.0001
Never a smoker	532,502 (68.7)	376,687 (58.4)	225,317 (46.5)	
Ex-smoker	105,378 (13.6)	118,655 (18.4)	98,980 (20.4)	
Current smoker	137,133 (17.7)	149,721 (23.2)	160,003 (33.0)	
Smoking status				<0.0001
Never a smoker	532,502 (68.7)	376,687 (58.4)	225,317 (46.5)	
Ex-smoker, <20 PYs	60,379 (7.8)	63,266 (9.8)	53,257 (11.0)	
Ex-smoker, ≥20 PYs	44,999 (5.8)	55,389 (8.6)	45,723 (9.4)	
Current smoker, <20 PYs	69,736 (9.0)	69,395 (10.8)	80,970 (16.7)	
Current smoker, ≥20 PYs	67,397 (8.7)	80,326 (12.5)	79,033 (16.3)	
Smoking, PYs				<0.0001
Non-smoking	532,502 (68.7)	376,687 (58.4)	225,317 (46.5)	
<5	34,895 (4.5)	33,921 (5.3)	35,016 (7.2)	
5–<10	35,249 (4.6)	37,576 (5.8)	38,919 (8.0)	
10–<15	29,394 (3.8)	33,564 (5.2)	34,304 (7.1)	
15–<20	112,396 (14.5)	135,715 (21.0)	124,756 (25.8)	
≥20	30,577 (4.0)	27,600 (4.3)	25,988 (5.4)	
Alcohol consumption				<0.0001
Non-drinker	554,456 (71.5)	402,511 (62.4)	234,504 (48.4)	
Mild drinker	220,557 (28.5)	242,552 (37.6)	249,796 (51.6)	
Physical activity, regular	169,425 (21.9)	131,456 (20.4)	85,375 (17.6)	<0.0001
Height (cm)	159.8 ± 9.1	161.9 ± 9.2	164.9 ± 9.2	<0.0001
Weight (kg)	58.2 ± 8.3	66.8 ± 8.4	76.8 ± 11.2	<0.0001
Waist circumference (cm)	78.9 ± 6.4	86.6 ± 5.6	93.3 ± 7.4	<0.0001
BMI, mean, kg/m^2^	22.7 ± 2.3	25.5 ± 2.3	28.2 ± 3.3	<0.0001
BMI, kg/m^2^				<0.0001
<18.5	28,320 (3.7)	1040 (0.2)	233 (0.1)	
18.5–<23	384,097 (49.6)	78,952 (12.2)	14,592 (3.0)	
23–<25	234,558 (30.3)	189,549 (29.4)	51,698 (10.7)	
25–<30	126,973 (16.4)	354,866 (55.0)	293,348 (60.6)	
≥30	1065 (0.1)	20,656 (3.2)	124,429 (25.7)	
Hypertension	376,551 (48.6)	382,564 (59.3)	305,818 (63.2)	<0.0001
Dyslipidemia	281,931 (36.4)	284,194 (44.1)	235,093 (48.5)	<0.0001
DM (≥5 years)	278,484 (35.9)	198,978 (30.9)	103,048 (21.3)	<0.0001
Insulin	70,165 (9.1)	45,772 (7.1)	25,515 (5.3)	<0.0001
OHA (≥3 drugs)	116,036 (15.0)	86,454 (13.4)	51,357 (10.6)	<0.0001
SBP, mmHg	126.1 ± 15.8	129.8 ± 15.5	132.4 ± 15.6	<0.0001
DBP, mmHg	76.6 ± 9.9	79.3 ± 9.9	82.0 ± 10.3	<0.0001
Glucose, mg/dL	140.6 ± 46.4	144.5 ± 45.9	151.5 ± 47.8	<0.0001
GFR, mL/min/1.73 m^2^	84.8 ±35.1	83.5 ± 34.5	85.2 ± 37.9	<0.0001
Total cholesterol, mg/dL	188.9 ± 39.9	199.2 ± 41.7	209.6 ± 44.1	<0.0001
HDL, mg/dL	54.4 ± 21.6	50.5 ± 21.7	48.7 ± 21.8	<0.0001
LDL, mg/dL	112.7 ± 37.8	115.2 ± 40.6	111.2 ± 45.1	<0.0001
* TG, mg/dL.	101.2 (101.1–101.3)	156.8 (156.7–157.0)	228.1 (227.8–228.4)	<0.0001

* Values are presented as geometric mean (95% CI). FLI, fatty liver index; PYs, pack-years; BMI, body mass index; DM, diabetes mellitus; OHA, oral hypoglycemic agent; SBP, systolic blood pressure; DBP, diastolic blood pressure; GFR, glomerular filtration rate; HDL, high-density lipoprotein; LDL, low-density lipoprotein; TG, triglyceride. The chi-square test was used to evaluate differences in the proportion of categorical variables. The ANOVA was used to determine differences between the means of continuous variables.

**Table 2 cancers-15-01209-t002:** Association between fatty liver and risk of HNC in DM patients.

Cancer Subtypes	FLI	Number of Subjects (%)	Cases	Duration	IR	Model 1	Model 2	Model 3
				(Person-Years)		aHR ^†^ (95% CI)	aHR ^‡^ (95% CI)	aHR ^§^ (95% CI)
Oral cavity	<30	775,013 (40.7)	340	5,328,939	6.38	1 (Ref.)	1 (Ref.)	1 (Ref.)
	<60	645,063 (33.9)	361	4,484,839	8.05	1.28 (1.08–1.51)	1.22 (1.03–1.44)	1.24 (1.05–1.46)
	≥60	484,300 (25.4)	248	3,352,292	7.4	1.42 (1.14–1.76)	1.29 (1.03–1.61)	1.33 (1.06–1.66)
Pharynx	<30	775,013 (40.7)	419	5,328,800	7.86	1 (Ref.)	1 (Ref.)	1 (Ref.)
	<60	645,063 (33.9)	435	4,484,675	9.7	1.34 (1.15–1.56)	1.28 (1.10–1.49)	1.30 (1.12–1.51)
	≥60	484,300 (25.4)	298	3,352,097	8.89	1.56 (1.28–1.89)	1.43 (1.18–1.75)	1.46 (1.20–1.79)
Larynx	<30	775,013 (40.7)	417	5,328,558	7.83	1 (Ref.)	1 (Ref.)	1 (Ref.)
	<60	645,063 (33.9)	378	4,484,698	8.43	1.18 (1.01–1.38)	1.09 (0.93–1.28)	1.11 (0.95–1.29)
	≥60	484,300 (25.4)	290	3,352,073	8.65	1.61 (1.33–1.96)	1.38 (1.13–1.69)	1.42 (1.16–1.74)
Salivary gland	<30	775,013 (40.7)	138	5,329,324	2.59	1 (Ref.)	1 (Ref.)	1 (Ref.)
	<60	645,063 (33.9)	132	4,485,368	2.94	1.01 (0.78–1.32)	1.02 (0.78–1.34)	1.05 (0.80–1.37)
	≥60	484,300 (25.4)	87	3,352,699	2.59	0.90 (0.63–1.20)	0.92 (0.64–1.33)	0.97 (0.67–1.41)

HNC, head and neck cancer; DM, diabetes mellitus; FLI, fatty liver index; IR, incidence rate (per 100,000 person-years); HR, hazard ratio; aHR, adjusted hazard ratio; CI, confidence interval; BMI, body mass index. ^†^ Adjusted for age, sex, and BMI. ^‡^ Adjusted for age, sex, BMI, smoking, alcohol consumption, regular physical activity, and income. ^§^ Adjusted for age, sex, BMI, smoking, alcohol consumption, regular physical activity, income, glucose, insulin, oral hypoglycemic agent, and DM duration.

**Table 3 cancers-15-01209-t003:** The aHR and 95% CI for HNC in DM patients based on the combination of obesity and fatty liver.

Cancer Subtypes	Subtype	FLI	Number of Subjects (%)	Cases	Duration	IR	aHR (95% CI)	*p*-Value
					(Person-Years)			
Oral cavity	Non-Obese (BMI < 25)	<30	646,975 (34.0)	292	4,427,638	6.6	1 (Ref.)	0.04
		<60	269,541 (14.2)	171	1,855,863	9.21	1.27 (1.05–1.54)	
		≥60	66,523 (3.5)	46	454,504	10.12	1.46 (1.06–2.00)	
	Obese (BMI ≥ 25)	<30	128,038 (6.7)	48	901,301	5.33	0.93 (0.68–1.26)	
		<60	375,522 (19.7)	190	2,628,976	7.23	1.14 (0.95–1.37)	
		≥60	417,777 (21.9)	202	2,897,788	6.97	1.22 (1.01–1.47)	
Pharynx	Non-Obese (BMI < 25)	<30	646,975 (34.0)	376	4,427,475	8.49	1 (Ref.)	0.009
		<60	269,541 (14.2)	242	1,855,684	13.04	1.19 (1.01–1.41)	
		≥60	66,523 (3.5)	66	454,470	14.52	1.25 (0.96–1.63)	
	Obese (BMI ≥ 25)	<30	128,038 (6.7)	43	901,325	4.77	0.78 (0.57–1.08)	
		<60	375,522 (19.7)	193	2,628,991	7.34	0.91 (0.76–1.08)	
		≥60	417,777 (21.9)	232	2,897,627	8.01	0.94 (0.80–1.12)	
Larynx	Non-Obese (BMI < 25)	<30	646,975 (34.0)	380	4,427,251	8.58	1 (Ref.)	<0.001
		<60	269,541 (14.2)	233	1,855,635	12.56	1.08 (0.91–1.27)	
		≥60	66,523 (3.5)	75	454,440	16.5	1.32 (1.03–1.70)	
	Obese (BMI ≥ 25)	<30	128,038 (6.7)	37	901,307	4.11	0.80 (0.57–1.12)	
		<60	375,522 (19.7)	145	2,629,063	5.52	0.72 (0.59–0.87)	
		≥60	417,777 (21.9)	215	2,897,633	7.42	0.91 (0.77–1.08)	
Salivary gland	Non-Obese (BMI < 25)	<30	646,975 (34.0)	109	4,427,975	2.46	1 (Ref.)	0.345
		<60	269,541 (14.2)	62	1,856,110	3.34	1.27 (0.92–1.74)	
		≥60	66,523 (3.5)	9	454,605	1.98	0.80 (0.40–1.60)	
	Obese (BMI ≥ 25)	<30	128,038 (6.7)	29	901,348	3.22	1.45 (0.96–2.21)	
		<60	375,522 (19.7)	70	2,629,258	2.66	1.10 (0.82–1.49)	
		≥60	417,777 (21.9)	78	2,898,094	2.69	1.20 (0.89–1.62)	

IR, incidence rate (per 100,000 person-years); aHR, adjusted hazard ratio; CI, confidence interval; HNC, head and neck cancer; DM, diabetes mellitus; FLI, fatty liver index; BMI, body mass index. Adjusted for age, sex, BMI, smoking, alcohol consumption, regular physical activity, income, glucose, insulin, oral hypoglycemic agent, and DM duration.

**Table 4 cancers-15-01209-t004:** Adjusted hazard ratio (aHR) and 95% confidence interval (CI) for head and neck cancer in DM patients according to the combination of age and fatty liver.

Cancer Subtypes	Subtype	FLI	Number of Subjects (%)	Cases	Duration	IR	aHR (95% CI)	*p*-Interaction
					(Person-Years)			
Oral cavity	Age < 40	<30	56,902 (3.0)	3	398,901	0.75	1 (Ref.)	0.826
		<60	35,815 (1.9)	5	251,349	1.99	2.20 (0.53–9.20)	
		≥60	57,353 (3.0)	10	399,913	2.5	2.74 (0.74–10.06)	
	Age 40–64	<30	439,334 (23.1)	145	3,102,295	4.67	1 (Ref.)	
		<60	396,895 (20.8)	173	2,801,002	6.18	1.21 (0.96–1.53)	
		≥60	325,215 (17.1)	151	2,270,146	6.65	1.34 (1.02–1.76)	
	Age ≥ 65	<30	278,777 (14.6)	192	1,827,742	10.51	1 (Ref.)	
		<60	212,353 (11.2)	183	1,432,487	12.78	1.25 (1.00–1.55)	
		≥60	101,732 (5.3)	87	682,233	12.75	1.26 (0.94–1.70)	
Pharynx	Age < 40	<30	56,902 (3.0)	6	398,889	1.5	1 (Ref.)	0.1437
		<60	35,815 (1.9)	3	251,354	1.19	0.75 (0.19–3.01)	
		≥60	57,353 (3.0)	8	399,898	2	1.55 (0.53–4.52)	
	Age 40–64	<30	439,334 (23.1)	219	3,102,126	7.06	1 (Ref.)	
		<60	396,895 (20.8)	230	2,800,875	8.21	1.11 (0.91–1.35)	
		≥60	325,215 (17.1)	194	2,269,984	8.55	1.28 (1.02–1.62)	
	Age ≥ 65	<30	278,777 (14.6)	194	1,827,785	10.61	1 (Ref.)	
		<60	212,353 (11.2)	202	1,432,447	14.1	1.53 (1.24–1.89)	
		≥60	101,732 (5.3)	96	682,215	14.07	1.69 (1.28–2.22)	
Larynx	Age < 40	<30	56,902 (3.0)	1	398,908	0.25	1 (Ref.)	0.1333
		<60	35,815 (1.9)	1	251,358	0.4	1.36 (0.09–21.64)	
		≥60	57,353 (3.0)	4	399,924	1	4.06 (0.45–36.35)	
	Age 40–64	<30	439,334 (23.1)	189	3,102,164	6.09	1 (Ref.)	
		<60	396,895 (20.8)	174	2,800,947	6.21	0.91 (0.74–1.14)	
		≥60	325,215 (17.1)	184	2,270,027	8.11	1.31 (1.03–1.66)	
	Age ≥ 65	<30	278,777 (14.6)	227	1,827,486	12.42	1 (Ref.)	
		<60	212,353 (11.2)	203	1,432,393	14.17	1.29 (1.05–1.58)	
		≥60	101,732 (5.3)	102	682,123	14.95	1.46 (1.12–1.91)	
Salivary gland	Age < 40	<30	56,902 (3.0)	5	398,895	1.25	1 (Ref.)	0.1517
		<60	35,815 (1.9)	1	251,354	0.4	0.25 (0.03–2.18)	
		≥60	57,353 (3.0)	3	399,929	0.75	0.44 (0.10–1.90)	
	Age 40–64	<30	439,334 (23.1)	82	3,102,424	2.64	1 (Ref.)	
		<60	396,895 (20.8)	71	2,801,241	2.53	0.85 (0.61–1.20)	
		≥60	325,215 (17.1)	61	2,270,393	2.69	0.89 (0.59–1.35)	
	Age ≥ 65	<30	278,777 (14.6)	51	1,828,005	2.79	1 (Ref.)	
		<60	212,353 (11.2)	60	1,432,772	4.19	1.44 (0.97–2.14)	
		≥60	101,732 (5.3)	23	682,377	3.37	1.13 (0.65–1.95)	

IR, incidence rate (per 100,000 person-years); aHR, adjusted hazard ratio; CI, confidence interval. Adjusted for age, sex, BMI, smoking, alcohol consumption, regular physical activity, income, glucose, insulin, oral hypoglycemic agent, and diabetes mellitus duration.

## Data Availability

The data that support the findings of this study are available from the Korean National Health Insurance Service (KNHIS) and were used under license for the current study (http://nhiss.nhis.or.kr, accessed on 10 February 2023). Restrictions apply to their availability (the data are not publicly available). The data are available with permission of the KNHIS from the authors upon reasonable request.

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
