# Peer review of "Fatty Liver and Risk of Head and Neck Cancer in Type 2 Diabetes Mellitus: A Nationwide Cohort Study"

_cancers, 2023, doi:10.3390/cancers15041209_

Round 1
Reviewer 1 Report
The authors addressed a very hot topic by performed a population-based retrospective cohort study aimed to investigate the association between non-alcoholic fatty liver disease (NAFLD) and the risk of head and neck cancer (HNC), separately based on cancer site, in a large cohort of patients affected by diabetes mellitus.
The work is original, results are novel and firstly demonstrate the association between NAFLD and HNC in diabetes patients. However, some criticisms remain tyo be addressed in order to improve the manuscript quality.
Major points
1. The presence of patients having less than 40 years and patients requiring insulin administration, suggests that a not negligeable percentage of patienst in the examined cohort were affected by type 1 diabetes mellitus (T1DM). The relative percentage of T1DM and type 2 diabetes mellitus (T2DM) should be shown in Table 1. Moreover, given the different pathophysiological mechanisms involved in the development and progression of T1DM and T2DM, the difference, if any, in the incidence of HNC between patients with T1DM and T2DM should be highlighted and commented.
2. The general characteristics (same of Table 1) of patients affected by HNC should be shown in a new Table and commented in Results.
3. What was the incidence of HN cancer in patients having less than 40 years?
4. The homeostasis model assessment: insulin resistance HOMA index should be calculated and shown in Table 1 as well as in the new table showing the charcteristics of DM patients affected by HNC.
5. Incidence rate and hazard ratios for the risk of HNC by each component of FLI, i.e. triglicerydes, waist circumference, body mass index, and gamma-glutamyl-transpeptidase, should be analysed and reported in the paper.
6. An association between NAFLD and esophageal cancer was recently reported by Lee and co-workers (PLOS One 2020 doi: 10.1371/journal.pone.0226351). However, this paper was not cited neither commented by the authors.
7. The reference list should be refreshed/updated by including a higher number of papers published between 2020 and 2022.
Author Response
Response to comments from the Reviewer 1
The authors addressed a very hot topic by performed a population-based retrospective cohort study aimed to investigate the association between non-alcoholic fatty liver disease (NAFLD) and the risk of head and neck cancer (HNC), separately based on cancer site, in a large cohort of patients affected by diabetes mellitus.
The work is original, results are novel and firstly demonstrate the association between NAFLD and HNC in diabetes patients. However, some criticisms remain to be addressed in order to improve the manuscript quality.
Major points
Comment #1. The presence of patients having less than 40 years and patients requiring insulin administration, suggests that a not negligeable percentage of patients in the examined cohort were affected by type 1 diabetes mellitus (T1DM). The relative percentage of T1DM and type 2 diabetes mellitus (T2DM) should be shown in Table 1.
Response: Thank you for the insightful comment. T2DM was defined as at least one claim per year based on ICD-10 codes of E11–14 with a prescription history of anti-diabetic medication or fasting glucose level ≥ 126 mg/dL at the health screening examination in our study. As type I DM is coded as E10 by ICD 10, T1DM is excluded in our study. This operational definition of Type 2 DM followed the several previous known studies (Ko et al.,2018; Hong et al.,2022; Yang et al., 2021; Jung et al., 2021; Bae et al., 2022). We clarified this in the METHODS section by adding the following sentence: “Type 1 DM was not included.”
[Materials and Methods] (page 3, paragraph 1, line 102)
T2DM was defined as at least one claim per year based on ICD-10 codes of E11–14 with a prescription history of antidiabetic medication or fasting glucose level ≥ 126 mg/dL at the health screening examination. Type 1 DM was not included.
[References]
Ko SH, Han K, Lee YH, et al. Past and Current Status of Adult Type 2 Diabetes Mellitus Management in Korea: A National Health Insurance Service Database Anlysis. Diabetes Metab J. 2018;42(2):93-100.
Hong YH, Chung IH, Han K, Chung S; Taskforce Team of Obesity Fact Sheet of the Korean Society for the Study of Obesity. Prevalence of Type 2 Diabetes Mellitus among Korean Children, Adolescents, and Adults Younger than 30 years: Changes from 2002 to 2016. Diabetes Metab J. 2022;46(2):297-306.
Yang YS, Han K, Sohn TS, Kim NH. Young-onset type 2 diabetes in South Korea: a review of the cuurent status and unmet need. Korean J Intern Med. 2021;36(5):1049-1058.
Jung CH, Son JW, Kang S, et al. Diabetes Fact Sheets in Korea, 2020: An Appraisal of Current Status. Diabetes Metab J. 2021;45(1):1-10.
Bae JH, Han KD, Ko SH et al. Diabetes Fact Sheet in Korea 2021. Diabetes Metab J. 2022;46(3):417-426.
Comment #2. Moreover, given the different pathophysiological mechanisms involved in the development and progression of T1DM and T2DM, the difference, if any, in the incidence of HNC between patients with T1DM and T2DM should be highlighted and commented.
Response: We appreciate your thoughtful comments and suggestions. As the reviewer noted, incidence of HNC between patients with T1DM and T2DM might be different, due to the different pathophysiological mechanisms of T1DM and T2DM in the development of cancer.
The current meta-analysis revealed that type 2 DM have an approximately 15% increased risk of oral cancer development (Stott-Miller et al., 2012). A possible explanation is hyperinsulinemia which is a potent growth factor that promote proliferation and carcinogenesis through insulin-like growth factor-1 (IGF-1) from various epithelial tumors including oral cancer (Novosyadlyy et al., 2010; Yen et al., 2014). Furthermore, since cancer cells rely on increased glucose consumption, hyperglycemia may promote the spread of carcinomas as well (Lorenzi et al., 1986; Liu et al., 2015). It could directly promote tumors and tissue damage due to the Glut-1 transporter (Ujpál et al., 2004). The rate of Glut-1 expression, indeed, correlates significantly with higher mortality and lower life expectancy of oral cancer patients (Kunkel et al., 2003).
However, evidence for head and neck cancers for type 1 diabetes, which is characterized by insulin deficiency, is currently very limited (Sona et al., 2018; Carstensen et al., 2016; Harding et al., 2015), probably due to smaller number of T1DM patients than T2DM patients and lower incidence of cancer in younger population.
As we responded to the previous comment, T1DM is excluded in our study, and therefore it is not possible to compare the incidence of HNC between patients with T1DM and T2DM in our study. We believe that future researches on T1DM and HNC could uncover the pathophysiological mechanisms. We added inclusion of T2DM patients as one of the limitation of this study as follows: “Fourth, we could not compare the incidence of HND between Type 1 DM and T2DM patients, as we included T2DM only. It is possible that two different types of DM may have different association with HNC risk, given the different pathophysiological mechanism between them.”
[Discussion] (page 13, paragraph 4, line 346)
Fourth, we could not compare the incidence of HNC between Type 1 DM and T2DM patients, as we included T2DM only. It is possible that two different types of DM may have different association with HNC risk, given the different pathophysiological mechanism between them.
[References]
Stott-Miller M, Chen C, Chuang SC, et al. History of diabetes and risk of head and neck cancer: a pooled analysis from the international head and neck cancer epidemiology consortium. Cancer Epidemiol Biomarkers Prev. Feb 2012;21(2):294-304.
Novosyadlyy R, LeRoith D. Hyperinsulinemia and type 2 diabetes: impact on cancer. Cell Cycle. Apr 15 2010;9(8):1449-50.
Yen YC, Shiah SG, Chu HC, et al. Reciprocal regulation of microRNA-99a and insulin-like growth factor I receptor signaling in oral squamous cell carcinoma cells. Mol Cancer. Jan 10 2014;13:6.
Lorenzi M, Montisano DF, Toledo S, Barrieux A. High glucose induces DNA damage in cultured human endothelial cells. J Clin Invest. Jan 1986;77(1):322-5.
Liu CJ, Chang WJ, Chen CY, et al. Dynamic cellular and molecular modulations of diabetes mediated head and neck carcinogenesis. Oncotarget. Oct 6 2015;6(30):29268-84.
Ujpál M, Matos O, Bíbok G, Somogyi A, Szabó G, Suba Z. Diabetes and oral tumors in Hungary: epidemiological correlations. Diabetes Care. Mar 2004;27(3):770-4.
Kunkel M, Reichert TE, Benz P, et al. Overexpression of Glut-1 and increased glucose metabolism in tumors are associated with a poor prognosis in patients with oral squamous cell carcinoma. Cancer. Feb 15 2003;97(4):1015-24.
Sona MF, Myung SK, Park K, Jargalsaikhan G. Type 1 diabetes mellitus and risk of cancer: a meta-analysis of observational studies. Jpn J Clin Oncol. 2018; 48(5):426-433.
Carstensen B, Read SH, Friis S, et al. Cancer incidence in persons with type 1 diabetes: a five-country study of 9,000 cancer in type 1 diabetic individuals. Diabetologia. 2016;59(5):980-988.
Harding JL, Shaw JE, Peeters A, Cartensen B, Magliano DJ. Cancer risk among people with type 1 and type 2 diabetes: disentangling tru associations, detection bias, and reverse causation. Diabetes Care. 2015; 38(2):264-270.
Comment #3. The general characteristics (same of Table 1) of patients affected by HNC should be shown in a new Table and commented in Results.
Response: We greatly appreciate the suggestions. Approximately 0.2% of study participants had head and neck cancer. The mean age in this group was older (62.8 ± 10.0 years) and more likely to be current smoker and alcohol consumer. We newly added the general characteristics of patients based on the head and neck cancer in supplementary table 1.
[Results] (page 4, paragraph 3, line 180)
Study participants who developed HNC were older and more likely to be current smoker and alcohol consumer (Supplementary table 1).
Supplementary Table 1. General characteristics of the subjects according to the risk of Head and Neck Cancer among type 2 diabetes mellitus patients.
|
Any Head and Neck Cancer |
|
|
Variables |
No |
Yes |
p-value |
Total, n (%) |
1,901,258 (99.8) |
3,118 (0.2) |
|
Age, years, mean ± SD |
57.6 ± 12.5 |
62.8 ± 10.0 |
<0.001 |
Age, years |
|
|
<0.001 |
<30 y |
28,786 (1.5) |
3 (0.1) |
|
30 ~ <40 y |
121,239 (6.4) |
42 (1.4) |
|
40 ~ <50 y |
336,969 (17.7) |
246 (7.9) |
|
50 ~ <60 y |
535,297 (28.2) |
815 (26.1) |
|
60 ~ <70 y |
507,174 (26.7) |
1,140 (36.6) |
|
70 ~ <80 y |
315,823 (16.6) |
755 (24.2) |
|
≥ 80 y |
55,970 (2.9) |
117 (3.8) |
|
Sex (male) |
1,070,160 (56.3) |
2,501 (80.2) |
<0.001 |
Income (lowest quartile) |
447,219 (23.5) |
753 (24.2) |
0.409 |
Smoking status |
|
|
<0.001 |
Never smoker |
1,133,225 (59.6) |
1,281 (41.1) |
|
Ex-smoker |
32,2278 (17.0) |
735 (23.6) |
|
Current smoker |
445,755 (23.5) |
1,102 (35.3) |
|
Smoking status |
|
|
<0.001 |
Never smoker |
1,133,225 (59.6) |
1,281 (41.1) |
|
Ex-smoker, < 20 PY |
176,594 (9.3) |
308 (9.9) |
|
Ex-smoker, ≥ 20 PY |
145,684 (7.7) |
427 (13.7) |
|
Current smoker, < 20 PY |
219,761 (11.6) |
340 (10.9) |
|
Current smoker, ≥ 20 PY |
225,994 (11.9) |
762 (24.4) |
|
Smoking, PY |
|
|
<0.001 |
Non-smoking |
1,133,225 (59.6) |
1,281 (41.1) |
|
<5 PY |
103,700 (5.5) |
132 (4.2) |
|
5 ~ <10 PY |
111,549 (5.9) |
195 (6.3) |
|
10 ~ <15 PY |
97,056 (5.1) |
206 (6.6) |
|
15 ~ <20 PY |
371,678 (19.6) |
1,189 (38.1) |
|
≥20 PY |
84,050 (4.4) |
115 (3.7) |
|
Alcohol consumption |
|
|
<0.001 |
Non-drinker |
1,189,761 (62.6) |
1,710 (54.8) |
|
Mild drinker |
711,497 (37.4) |
1,408 (45.2) |
|
Physical activity. regular |
385,559 (20.3) |
697 (22.4) |
0.004 |
Height (cm) |
161.8 ± 9.4 |
164.0 ± 7.9 |
<0.001 |
Weight (kg) |
65.8 ± 11.8 |
66.2 ± 10.5 |
0.112 |
Waist Circumference (cm) |
85.1 ± 8.7 |
86.3 ± 8.2 |
<0.001 |
BMI, mean, kg/m2 |
25.1 ± 3.4 |
24.6 ± 3.2 |
<0.001 |
BMI, kg/m2 |
|
|
<0.001 |
<18.5 |
29,519 (1.6) |
74 (2.4) |
|
18.5 ~ <23 |
476,772 (25.1) |
869 (27.9) |
|
23 ~ <25 |
474,940 (25.0) |
865 (27.7) |
|
25 ~ <30 |
774,033 (40.7) |
1,154 (37.0) |
|
≥30 |
145,994 (7.7) |
156 (5.0) |
|
Hypertension |
1,062,940 (55.9) |
1,993 (63.9) |
<0.001 |
Dyslipidemia |
799,961 (42.1) |
1,257 (40.3) |
0.047 |
DM (≥5 years) |
579,379 (30.5) |
1,131 (36.3) |
<0.001 |
Insulin |
141,188 (7.4) |
264 (8.5) |
0.027 |
OHA (≥ 3 drugs) |
253,334 (13.3) |
513 (16.5) |
<0.001 |
SBP, mmHg |
128.9 ± 15.9 |
130.8 ± 16.2 |
<0.001 |
DBP, mmHg |
78.9 ± 10.2 |
79.1 ± 10.4 |
0.331 |
Glucose, mg/dL |
144.7 ± 46.8 |
141.5 ± 45.2 |
<0.001 |
GFR, mL/min/1.73m2 |
84.5 ± 35.7 |
82.5 ± 40.9 |
0.003 |
Total cholesterol, mg/Dl |
197.7 ± 42.4 |
192.9 ± 41.9 |
<0.001 |
HDL, mg/dL |
51.6 ± 21.8 |
51.1 ± 24.6 |
0.167 |
LDL, mg/dL |
113.2 ± 40.7 |
109.4 ± 45.0 |
<0.001 |
*TG, mg/dL |
144.3 (144.2–144.4) |
144.8 (141.9–147.7) |
0.758 |
* Values presented as Geometric mean (95% Confidence Interval)
FLI, fatty liver index; PY, pack-years; BMI, body mass index; DM, diabetes mellitus; OHA, oral hypoglycemic agent; SBP, Systolic Blood Pressure; DBP, Diastolic Blood Pressure ;GFR, glomerular filtration rate; HDL, high density lipoprotein cholesterol; LDL, low density lipoprotein cholesterol; TG, triglyceride.
Comment #4. What was the incidence of HN cancer in patients having less than 40 years?.
Response: We greatly appreciate the suggestions. Only 18, 17, 6, and 9 cases of oral, pharynx, larynx, and salivary gland cancers occurred in those who were under 40 years old at baseline. The incidence rates (per 100,000 person-years) of oral, pharynx, larynx, and salivary gland cancers were only 2.50, 2.00, 1.00, and 0.75 respectively with NAFLD under 40 years old. We newly added a table about incidence and risk of head and neck cancer stratified by age in Table 4. There was no significant interaction across the age groups.
[Results] (page 6, paragraph 3, line 217)
Table 4 shows the association between the risk of HNC and combination of age and FLI. There was no significant interaction across the age groups
Table 4. Adjusted Hazard ratio (aHR) and 95% confidence interval (CI) for head and neck cancer in DM patients according to the combination of Age and fatty liver.
Cancer subtypes |
Subtype |
FLI |
Number of subjects (%) |
Cases |
Duration (person-years) |
IR |
aHR (95% CI) |
p-interaction |
Oral cavity |
Age < 40 |
< 30 |
56,902 (3.0) |
3 |
398,901 |
0.75 |
1 (Ref.) |
0.826 |
< 60 |
35,815 (1.9) |
5 |
251,349 |
1.99 |
2.20 (0.53–9.20) |
|
||
≥ 60 |
57,353 (3.0) |
10 |
399,913 |
2.50 |
2.74 (0.74–10.06) |
|
||
Age 40-64 |
< 30 |
439,334 (23.1) |
145 |
3,102,295 |
4.67 |
1 (Ref.) |
|
|
< 60 |
396,895 (20.8) |
173 |
2,801,002 |
6.18 |
1.21 (0.96–1.53) |
|
||
≥ 60 |
325,215 (17.1) |
151 |
2,270,146 |
6.65 |
1.34 (1.02–1.76) |
|
||
Age ≥ 65 |
< 30 |
278,777 (14.6) |
192 |
1,827,742 |
10.51 |
1 (Ref.) |
|
|
< 60 |
212,353 (11.2) |
183 |
1,432,487 |
12.78 |
1.25 (1.00–1.55) |
|
||
≥ 60 |
101,732 (5.3) |
87 |
682,233 |
12.75 |
1.26 (0.94–1.70) |
|
||
Pharynx |
Age < 40 |
< 30 |
56,902 (3.0) |
6 |
398,889 |
1.50 |
1 (Ref.) |
0.1437 |
< 60 |
35,815 (1.9) |
3 |
251,354 |
1.19 |
0.75 (0.19–3.01) |
|
||
≥ 60 |
57,353 (3.0) |
8 |
399,898 |
2.00 |
1.55 (0.53–4.52) |
|
||
Age 40-64 |
< 30 |
439,334 (23.1) |
219 |
3,102,126 |
7.06 |
1 (Ref.) |
|
|
< 60 |
396,895 (20.8) |
230 |
2,800,875 |
8.21 |
1.11 (0.91–1.35) |
|
||
≥ 60 |
325,215 (17.1) |
194 |
2,269,984 |
8.55 |
1.28 (1.02–1.62) |
|
||
Age ≥ 65 |
< 30 |
278,777 (14.6) |
194 |
1,827,785 |
10.61 |
1 (Ref.) |
|
|
< 60 |
212,353 (11.2) |
202 |
1,432,447 |
14.10 |
1.53 (1.24–1.89) |
|
||
≥ 60 |
101,732 (5.3) |
96 |
682,215 |
14.07 |
1.69 (1.28–2.22) |
|
||
Larynx |
Age < 40 |
< 30 |
56,902 (3.0) |
1 |
398,908 |
0.25 |
1 (Ref.) |
0.1333 |
< 60 |
35,815 (1.9) |
1 |
251,358 |
0.40 |
1.36 (0.09–21.64) |
|
||
≥ 60 |
57,353 (3.0) |
4 |
399,924 |
1.00 |
4.06 (0.45–36.35) |
|
||
Age 40-64 |
< 30 |
439,334 (23.1) |
189 |
3,102,164 |
6.09 |
1 (Ref.) |
|
|
< 60 |
396,895 (20.8) |
174 |
2,800,947 |
6.21 |
0.91 (0.74–1.14) |
|
||
≥ 60 |
325,215 (17.1) |
184 |
2,270,027 |
8.11 |
1.31 (1.03–1.66) |
|
||
Age ≥ 65 |
< 30 |
278,777 (14.6) |
227 |
1,827,486 |
12.42 |
1 (Ref.) |
|
|
< 60 |
212,353 (11.2) |
203 |
1,432,393 |
14.17 |
1.29 (1.05–1.58) |
|
||
≥ 60 |
101,732 (5.3) |
102 |
682,123 |
14.95 |
1.46 (1.12–1.91) |
|
||
Salivary gland |
Age < 40 |
< 30 |
56,902 (3.0) |
5 |
398,895 |
1.25 |
1 (Ref.) |
0.1517 |
< 60 |
35,815 (1.9) |
1 |
251,354 |
0.40 |
0.25 (0.03–2.18) |
|
||
≥ 60 |
57,353 (3.0) |
3 |
399,929 |
0.75 |
0.44 (0.10–1.90) |
|
||
Age 40-64 |
< 30 |
439,334 (23.1) |
82 |
3,102,424 |
2.64 |
1 (Ref.) |
|
|
< 60 |
396,895 (20.8) |
71 |
2,801,241 |
2.53 |
0.85 (0.61–1.20) |
|
||
≥ 60 |
325,215 (17.1) |
61 |
2,270,393 |
2.69 |
0.89 (0.59–1.35) |
|
||
Age ≥ 65 |
< 30 |
278,777 (14.6) |
51 |
1,828,005 |
2.79 |
1 (Ref.) |
|
|
< 60 |
212,353 (11.2) |
60 |
1,432,772 |
4.19 |
1.44 (0.97–2.14) |
|
||
≥ 60 |
101,732 (5.3) |
23 |
682,377 |
3.37 |
1.13 (0.65–1.95) |
|
IR, incidence rate (per 100,000 person-years); aHR, adjusted hazard ratio; CI, confidence interval.
Adjusted for age, sex, BMI, smoking, alcohol consumption, regular physical activity, income, glucose, insulin, oral hypoglycemic agent, and diabetes mellitus duration.
Comment #5. The homeostasis model assessment: insulin resistance HOMA index should be calculated and shown in Table 1 as well as in the new table showing the characteristics of DM patients affected by HNC
Response: Thank you for the comments. Unfortunately, we could not calculate the HOMA-IR index because the NHIS database does not contains data of insulin resistance such fasting insulin.
Comment #6. Incidence rate and hazard ratios for the risk of HNC by each component of FLI, i.e. triglicerydes, waist circumference, body mass index, and gamma-glutamyl-transpeptidase, should be analysed and reported in the paper.
Response: We appreciate the reviewer’s suggestion. We newly added table about association between fatty liver index components such as body mass index, waist circumference, triglyceride and gamma-glutamyl transferase and risk of head and neck cancer in supplementary table 2. These components were dichotomized for analyses according to the criteria to report abnormality of Korean national health screening program. Among these component, GGT was found as a better biomarker of liver fat and increased GGT was most prominently associated with increased HNC risk.
In previous study (Kozakova et al., 2012) which showed an association between fatty liver and carotid atherosclerosis, the author noted that GGT is an independently established factor possibly linking fatty liver and atherosclerosis when FLI in the model was replaced by variables used in its equation (e.g., body mass index, waist circumference, plasma triglycerides, and GGT). In addition, a meta-analysis study (Fraser et al., 2009) showed a potentially stronger association of GGT and diabetes. GGT is present on the surface of most cell types and is highly active in organs other than the liver, such as the kidney and pancreas (Hanigan et al.,1996). GGT is the enzyme responsible for the extracellular catabolism of antioxidant glutathione (Turgut et al, 2006) and may be linked to greater oxidative stress (Ceriello et al., 2003). Because oxidative stress has been implicated in insulin resistance, diabetes, and cardiovascular disease (Ceriello et al., 2003), GGT's potentially stronger association with diabetes may reflect its associations with several different processes relevant to diabetes pathogenesis. We added this additional analysis results in the DISCUSSION section.
[DISCUSSION] (page 13, paragraph 1, line 300)
In addition, among fatty liver index components, GGT was the most prominent biomarker of liver fat (Supplementary table 2). It is because GGT, an enzyme responsible for the extracellular catabolism of antioxidant glutathione may be linked to greater oxidative stress which has been implicated in insulin resistance and diabetes
[References]
Kozakova M, Palombo C, Eng MP, et al. Fatty liver index, gamma-glutamyltransferase, and early carotid plaques. Hepatology. 2012;55(5):1406-1415
Fraser A, Harris R, Sattar N, Ebrahim S, Davey Smith G, Lawlor DA. Alanine aminotransferase, gamma-glutamyltransferase, and incident diabetes: the British Women's Heart and Health Study and meta-analysis. Diabetes Care. 2009;32(4):741-750
Hanigan MH, Frierson HF, Jr: Immunohistochemical detection of gamma-glutamyl transpeptidase in normal human tissue. J Histochem Cytochem 44: 1101– 1108, 1996
Turgut O, Yilmaz A, Yalta K, Karadas F, Birhan Yilmaz M: Gamma-glutamyltransferase is a promising biomarker for cardiovascular risk. Med Hypotheses 67: 1060– 1064, 2006
Ceriello A, Motz E: Is oxidative stress the pathogenic mechanism underlying insulin resistance, diabetes, and cardiovascular disease? The common soil hypothesis revisited. Arterioscler Thromb Vasc Biol 24: 816– 823, 2004
Supplementary Table 2. Association between components of fatty liver index and risk of head and neck cancer in type 2 diabetes mellitus patients.
Cancer subtypes |
BMI (kg/m2) |
Number of subjects (%) |
Cases |
Duration (Person-years) |
IR |
Model 1 aHR1 (95% CI) |
Model 2 aHR2 (95% CI) |
Model 3 aHR3 (95% CI) |
Oral cavity |
< 25 |
983,039 (51.6) |
509 |
6,738,004 |
7.55 |
1 (Ref.) |
1 (Ref.) |
1 (Ref.) |
≥ 25 |
921,337 (48.4) |
440 |
6,428,065 |
6.85 |
0.91 (0.74–1.11) |
0.91 (0.74–1.11) |
0.91 (0.75–1.11) |
|
Pharynx |
< 25 |
983,039 (51.6) |
684 |
6,737,629 |
10.15 |
1 (Ref.) |
1 (Ref.) |
1 (Ref.) |
≥ 25 |
921,337 (48.4) |
468 |
6,427,943 |
7.28 |
1.04 (0.86–1.26) |
1.04 (0.86–1.25) |
1.04 (0.86–1.25) |
|
Larynx |
< 25 |
983,039 (51.6) |
688 |
6,737,326 |
10.21 |
1 (Ref.) |
1 (Ref.) |
1 (Ref.) |
≥ 25 |
921,337 (48.4) |
397 |
6,428,003 |
6.18 |
0.88 (0.72–1.06) |
0.87 (0.72–1.06) |
0.87 (0.72–1.06) |
|
Salivary gland |
< 25 |
983,039 (51.6) |
180 |
6,738,690 |
2.67 |
1 (Ref.) |
1 (Ref.) |
1 (Ref.) |
≥ 25 |
921,337 (48.4) |
177 |
6,428,701 |
2.75 |
1.08 (0.78–1.50) |
1.09 (0.79–1.51) |
1.10 (0.79–1.52) |
|
Cancer subtypes |
TG (mg/dL) |
|
|
|
|
|
|
|
Oral cavity |
< 150 |
1,024,618 (53.8) |
506 |
7,064,351 |
7.16 |
1 (Ref.) |
1 (Ref.) |
1 (Ref.) |
|
≥ 150 |
879,758 (46.2) |
443 |
6,101,719 |
7.26 |
1.09 (0.96–1.24) |
1.05 (0.92–1.19) |
1.06 (0.93–1.20) |
Pharynx |
< 150 |
1,024,618 (53.8) |
606 |
7,064,114 |
8.58 |
1 (Ref.) |
1 (Ref.) |
1 (Ref.) |
|
≥ 150 |
879,758 (46.2) |
546 |
6,101,459 |
8.95 |
1.15 (1.02–1.30) |
1.11 (0.98–1.25) |
1.12 (0.99–1.26) |
Larynx |
< 150 |
1,024,618 (53.8) |
571 |
7,063,902 |
8.08 |
1 (Ref.) |
1 (Ref.) |
1 (Ref.) |
|
≥ 150 |
879,758 (46.2) |
514 |
6,101,427 |
8.42 |
1.22 (1.08–1.38) |
1.13 (1.00–1.28) |
1.14 (1.01–1.29) |
Salivary gland |
< 150 |
1,024,618 (53.8) |
189 |
7,064,995 |
2.68 |
1 (Ref.) |
1 (Ref.) |
1 (Ref.) |
|
≥ 150 |
879,758 (46.2) |
168 |
6,102,396 |
2.75 |
1.05 (0.85–1.30) |
1.05 (0.85–1.30) |
1.07 (0.87–1.33) |
Cancer subtypes |
WC (cm) |
|
|
|
|
|
|
|
Oral cavity |
M<90,W<80 |
990,173 (51.2) |
467 |
6,815,946 |
6.85 |
1 (Ref.) |
1 (Ref.) |
1 (Ref.) |
|
M≥90,W≥80 |
914,203 (48.0) |
482 |
6,350,124 |
7.59 |
1.21 (1.02–1.42) |
1.19 (1.00–1.40) |
1.18 (1.00–1.39) |
Pharynx |
M<90,W<80 |
990,173 (51.2) |
705 |
6,815,372 |
10.34 |
1 (Ref.) |
1 (Ref.) |
1 (Ref.) |
|
M≥90,W≥80 |
914,203 (48.0) |
447 |
6,350,201 |
7.04 |
1.16 (1.00–1.36) |
1.14 (0.98–1.33) |
1.14 (0.98–1.33) |
Larynx |
M<90,W<80 |
990,173 (51.2) |
705 |
6,815,139 |
10.35 |
1 (Ref.) |
1 (Ref.) |
1 (Ref.) |
|
M≥90,W≥80 |
914,203 (48.0) |
380 |
6,350,190 |
5.98 |
1.12 (0.95–1.31) |
1.07 (0.91–1.26) |
1.07 (0.91–1.25) |
Salivary gland |
M<90,W<80 |
990,173 (51.2) |
175 |
6,816,594 |
2.57 |
1 (Ref.) |
1 (Ref.) |
1 (Ref.) |
|
M≥90,W≥80 |
914,203 (48.0) |
182 |
6,350,796 |
2.87 |
1.28 (0.97–1.68) |
1.27 (0.97–1.67) |
1.27 (0.96–1.66) |
Cancer subtypes |
GGT (IU/L) |
|
|
|
|
|
|
|
Oral cavity |
M<63,W<35 |
1,436,534 (75.4) |
682 |
9,949,374 |
6.85 |
1 (Ref.) |
1 (Ref.) |
1 (Ref.) |
|
M≥63,W≥35 |
467,842 (24.6) |
267 |
3,216,695 |
8.30 |
1.48 (1.28–1.71) |
1.41 (1.22–1.63) |
1.44 (1.24–1.68) |
Pharynx |
M<63,W<35 |
1,436,534 (75.4) |
850 |
9,948,896 |
8.54 |
1 (Ref.) |
1 (Ref.) |
1 (Ref.) |
|
M≥63,W≥35 |
467,842 (24.6) |
302 |
3,216,676 |
9.39 |
1.39 (1.21–1.59) |
1.33 (1.16–1.53) |
1.34 (1.17–1.54) |
Larynx |
M<63,W<35 |
1,436,534 (75.4) |
793 |
9,948,717 |
7.97 |
1 (Ref.) |
1 (Ref.) |
1 (Ref.) |
|
M≥63,W≥35 |
467,842 (24.6) |
292 |
3,216,613 |
9.08 |
1.58 (1.37–1.81) |
1.47 (1.28–1.70) |
1.50 (1.30–1.73) |
Salivary gland |
M<63,W<35 |
1,436,534 (75.4) |
290 |
9,950,222 |
2.91 |
1 (Ref.) |
1 (Ref.) |
1 (Ref.) |
|
M≥63,W≥35 |
467,842 (24.6) |
67 |
3,217,168 |
2.08 |
0.78 (0.60–1.02) |
0.80 (0.61–1.06) |
0.83 (0.63–1.10) |
M, male; F, female; BMI, body mass index; WC, waist circumference; TG, triglyceride. GGT, gamma-glutamyl-transferase; IR, incidence rate (per 100,000 person-years); HR, hazard ratio; aHR, adjusted hazard ratio; CI, confidence interval.
1Adjusted for age, sex, and BMI.
2Adjusted for age, sex, BMI, smoking, alcohol consumption, regular physical activity, and income.
3Adjusted for age, sex, BMI, smoking, alcohol consumption, regular physical activity, income, glucose, insulin, oral hypoglycemic agent, and diabetes mellitus duration.
Statistically significant values are marked in bold.
Comment #7. An association between NAFLD and esophageal cancer was recently reported by Lee and co-workers (PLOS One 2020 doi:10.1371/journal.pone.0226351). However, this paper was not cited neither commented by authors.
Response: We appreciate the reviewer’s thoughtful comments. We added the reference in the introduction section.
[Introduction] (page 2, paragraph 1, line 73)
With the increasing prevalence of NAFLD in parallel with diabetes, the possible association between NAFLD and cancer development was suggested in several previous studies. In addition, squamous cell carcinoma as well as adenocarcinoma were shown associated with NAFLD.
[References]
Lee JM, Park YM, Yun JS, et al. The association between nonalcoholic fatty liver disease and esophageal stomach, or colorectal cancer: National population-based cohort study. PLoS One. 2020;15(1):e0226351.
Comment #8. The reference list should be refreshed/updated by including a higher number of papers published between 2020 and 2022.
Response: We appreciate the reviewer’s suggestion. We updated the old references, such as published in 1998, to those published recently. The new added reference is highlighted in yellow.
[References]
Lee JM, Park YM, Yun JS, et al. The association between nonalcoholic fatty liver disease and esophageal stomach, or colorectal cancer: National population-based cohort study. PLoS One. 2020;15(1):e0226351.
Lee JE, Han K, Yoo J, et al. Association between Metabolic Syndrome and Risk of Esophageal Cancer: a Nationwide Population-based Study. Cancer Epidemiol biomarkers Prev. 2022;EPI-22-0703.
Pan SY, de Groh M, Morrison H. A Case-Control Study of Risk Factors for Salivary Gland Cancer in Canada. J Cancer Epidemiol. 2017;2017:4909214

Reviewer 2 Report
In this manuscript, the authors demonstrate that NAFLD was associated with an increased risk of developing HNC in the oral cavity, pharynx, and larynx but not in the salivary gland. The manuscript is well written; however, some clarifications would be useful.
Manuscript Concerns:
1. “The incidence of non-alcoholic fatty liver disease (NAFLD) is > 2-fold higher in patients with diabetes mellitus (DM)”, please add the reference which supports this conclusion.
2. Briefly the authors talk about diabetes (DM), but in line 40, they address T2DM, can they explain?
3. In line 48, “Obese patients showed a lower risk of cancer than non-obese patients.”, please list the supporting data, especially the significance.
4. In Table 1, how to understand the p-value, the author should list the details of how they calculate the results, including the groups and the methods.
5. In Table 3, how to understand there are fewer cases of cancer in Non-Obese (BMI<25) when the FLI is bigger, but there are more cases of cancer in Obese (BMI>25) when the FLI is bigger?
6. In Table 3, please let us know the mean of N (%).
7. In Table 3, please list the unit of “Duration”.
8. In figure 1, the quality is low, please change to a clearer one.
Author Response
Response to comments from the Reviewer 2
In this manuscript, the authors demonstrate that NAFLD was associated with an increased risk of developing HNC in the oral cavity, pharynx, and larynx but not in the salivary gland. The manuscript is well written; however, some clarifications would be useful.
Manuscript Concerns:
Comment #1. The incidence of non-alcoholic fatty liver disease (NAFLD) is > 2-fold higher in patients with diabetes mellitus (DM)”, please add the reference which supports this conclusion.
Response: We appreciate the reviewer’s thoughtful comments. We addressed the reference in the simple summary in line 29 and introduction in line 72 as follows: The global prevalence of non-alcoholic fatty liver disease (NAFLD) in DM is 55.5%, ≥ 2-fold higher than in the general population according to the systematic review and meta-analysis (Younossi et al.2019).
[References]
Younossi ZM, Golabi P, de Avila L, et al. The global epidemiology of NAFLD and NASH in patients with type 2 diabetes: A systematic review and meta-analysis. J Hepatol. 2019; 71(4):793-801.
Comment #2. Briefly the authors talk about diabetes (DM), but in line 40, they address T2DM, can they explain?
Response: Thank you for the insightful comment. As we explained at the response of Reviewer 1’s comment #1, our study only targeted the T2DM patients. Diabetes (DM) was meant T2DM in the study. All terms have changed from DM to T2DM to prevent the confusion.
Comment #3. In line 48, “Obese patients showed a lower risk of cancer than non-obese patients.”, please list the supporting data, especially the significance.
Response: Thank you for the grateful comments. We additionally analyzed to reveal the relationship between obesity and HNC (Supplementary table 2). We found that there was no association between obesity and HNC. Currently, the association between obesity and the risk of HNC remains controversial (Gaudet et al., 2015). Ward noted that central adiposity but not BMI was associated with HNC (Ward et al.2017). In 2018, the World Cancer Research Fund (WCRF) concluded that higher body fat showed significant positive association with HNC for never-smokers from a pooled analysis of 20 cohort studies (WCRF/AICR, 2007). Therefore, NAFLD appears to be a better predictor than BMI in reflecting an obesity phenotype with higher malignancy potential because it is closely associated with central adiposity and insulin resistance (Allen et al., 2019)
In fact, we found the pattern that obese patients showed lower risk of cancer than non-obese patients with the same level of FLI by analyzing patients classified into 6 group by combination of obesity and fatty liver index (Non-obese with FLI<30, non-obese with FLI<60, non-obese with FLI ≥60, obese with FLI<30, obese with FLI<60, obese with FLI ≥60) in Table 3. For example, the risk of larynx cancer was higher in the non-obese patients with FLI ≥60 (aHR 1.32, 95% CI 1.03–1.70) than obese patients with FLI ≥60 (aHR 0.91, 95% CI 0.77–1.08).
We changed the sentence in line 48 as follows to provide more detailed information and emphasize the importance. “There was no association between obesity and HNC. However, obese patients showed lower risk of cancer for oral cavity (p =0.040), pharynx (p =0.009), and larynx (p <0.001) than non-obese patients with the same FLI level.”
[Abstract] (page 1, paragraph 2, line 48)
There was no association between obesity and HNC. However, obese patients showed lower risk of cancer for oral cavity (p =0.040), pharynx (p =0.009), and larynx (p <0.001) than non-obese patients with the same FLI level
[Discussion] (page 12, paragraph 3, line 291)
In the present study, Obesity defined by BMI and HNC (oral cavity, pharynx, and larynx cancer) showed no association (Supplementary table 2). Currently, the association between obesity and the risk of HNC remains controversial. Ward et al. noted that central adiposity but not BMI was associated with HNC. In 2018, the World Cancer Research Fund (WCRF) concluded that higher body fat showed significant positive association with HNC for never-smokers from a pooled analysis of 20 cohort studies. NAFLD appears to be a better predictor than BMI in reflecting an obesity phenotype with higher malignancy potential because it is closely associated with central adiposity and insulin resistance. In addition, among fatty liver index components, GGT was the most prominent biomarker of liver fat (Supplementary table 2). It is because GGT, an enzyme responsible for the extracellular catabolism of antioxidant glutathione may be linked to greater oxidative stress which has been implicated in insulin resistance and diabetes. Further research to determine the relationship between NAFLD and HNC will aid in understanding the detailed mechanism of insulin resistance and lipogenesis in central adiposity.
[References]
Gaudet MM, Kitahara CM, Newton CC, et al. Anthropometry and head and neck cancer: a pooled analysis of cohort data. Int J Epidemiol. 2015;44(2):673-681.
Ward HA, Wark PA, Muller DC, et al. Measured Adiposity in Relation to Head and Neck Cancer Risk in the European Prospective Investigation into Cancer and Nutrition. Cancer Epidemiol Biomarkers Prev. 2017;26(6):895-904.
The World Cancer Research Fund/American Institute for Cancer Research. Food, Nutrition, Physical Activity and the Prevention of Cancer: a Global Perspective. 2007
Allen AM, Hicks SB, Mara KC, Larson JJ, Therneau TM. The risk of incident extrahepatic cancers is higher in non-alcoholic fatty liver disease than obesity – A longitudinal cohort study. J Hepatol. 2019;71(6):1229-1236.
Supplementary Table 2. Association between components of fatty liver index and risk of head and neck cancer in type 2 diabetes mellitus patients.
Cancer subtypes |
BMI (kg/m2) |
Number of subjects (%) |
Cases |
Duration (Person-years) |
IR |
Model 1 aHR1 (95% CI) |
Model 2 aHR2 (95% CI) |
Model 3 aHR3 (95% CI) |
Oral cavity |
< 25 |
983,039 (51.6) |
509 |
6,738,004 |
7.55 |
1 (Ref.) |
1 (Ref.) |
1 (Ref.) |
≥ 25 |
921,337 (48.4) |
440 |
6,428,065 |
6.85 |
0.91 (0.74–1.11) |
0.91 (0.74–1.11) |
0.91 (0.75–1.11) |
|
Pharynx |
< 25 |
983,039 (51.6) |
684 |
6,737,629 |
10.15 |
1 (Ref.) |
1 (Ref.) |
1 (Ref.) |
≥ 25 |
921,337 (48.4) |
468 |
6,427,943 |
7.28 |
1.04 (0.86–1.26) |
1.04 (0.86–1.25) |
1.04 (0.86–1.25) |
|
Larynx |
< 25 |
983,039 (51.6) |
688 |
6,737,326 |
10.21 |
1 (Ref.) |
1 (Ref.) |
1 (Ref.) |
≥ 25 |
921,337 (48.4) |
397 |
6,428,003 |
6.18 |
0.88 (0.72–1.06) |
0.87 (0.72–1.06) |
0.87 (0.72–1.06) |
|
Salivary gland |
< 25 |
983,039 (51.6) |
180 |
6,738,690 |
2.67 |
1 (Ref.) |
1 (Ref.) |
1 (Ref.) |
≥ 25 |
921,337 (48.4) |
177 |
6,428,701 |
2.75 |
1.08 (0.78–1.50) |
1.09 (0.79–1.51) |
1.10 (0.79–1.52) |
|
Cancer subtypes |
TG (mg/dL) |
|
|
|
|
|
|
|
Oral cavity |
< 150 |
1,024,618 (53.8) |
506 |
7,064,351 |
7.16 |
1 (Ref.) |
1 (Ref.) |
1 (Ref.) |
|
≥ 150 |
879,758 (46.2) |
443 |
6,101,719 |
7.26 |
1.09 (0.96–1.24) |
1.05 (0.92–1.19) |
1.06 (0.93–1.20) |
Pharynx |
< 150 |
1,024,618 (53.8) |
606 |
7,064,114 |
8.58 |
1 (Ref.) |
1 (Ref.) |
1 (Ref.) |
|
≥ 150 |
879,758 (46.2) |
546 |
6,101,459 |
8.95 |
1.15 (1.02–1.30) |
1.11 (0.98–1.25) |
1.12 (0.99–1.26) |
Larynx |
< 150 |
1,024,618 (53.8) |
571 |
7,063,902 |
8.08 |
1 (Ref.) |
1 (Ref.) |
1 (Ref.) |
|
≥ 150 |
879,758 (46.2) |
514 |
6,101,427 |
8.42 |
1.22 (1.08–1.38) |
1.13 (1.00–1.28) |
1.14 (1.01–1.29) |
Salivary gland |
< 150 |
1,024,618 (53.8) |
189 |
7,064,995 |
2.68 |
1 (Ref.) |
1 (Ref.) |
1 (Ref.) |
|
≥ 150 |
879,758 (46.2) |
168 |
6,102,396 |
2.75 |
1.05 (0.85–1.30) |
1.05 (0.85–1.30) |
1.07 (0.87–1.33) |
Cancer subtypes |
WC (cm) |
|
|
|
|
|
|
|
Oral cavity |
M<90,W<80 |
990,173 (51.2) |
467 |
6,815,946 |
6.85 |
1 (Ref.) |
1 (Ref.) |
1 (Ref.) |
|
M≥90,W≥80 |
914,203 (48.0) |
482 |
6,350,124 |
7.59 |
1.21 (1.02–1.42) |
1.19 (1.00–1.40) |
1.18 (1.00–1.39) |
Pharynx |
M<90,W<80 |
990,173 (51.2) |
705 |
6,815,372 |
10.34 |
1 (Ref.) |
1 (Ref.) |
1 (Ref.) |
|
M≥90,W≥80 |
914,203 (48.0) |
447 |
6,350,201 |
7.04 |
1.16 (1.00–1.36) |
1.14 (0.98–1.33) |
1.14 (0.98–1.33) |
Larynx |
M<90,W<80 |
990,173 (51.2) |
705 |
6,815,139 |
10.35 |
1 (Ref.) |
1 (Ref.) |
1 (Ref.) |
|
M≥90,W≥80 |
914,203 (48.0) |
380 |
6,350,190 |
5.98 |
1.12 (0.95–1.31) |
1.07 (0.91–1.26) |
1.07 (0.91–1.25) |
Salivary gland |
M<90,W<80 |
990,173 (51.2) |
175 |
6,816,594 |
2.57 |
1 (Ref.) |
1 (Ref.) |
1 (Ref.) |
|
M≥90,W≥80 |
914,203 (48.0) |
182 |
6,350,796 |
2.87 |
1.28 (0.97–1.68) |
1.27 (0.97–1.67) |
1.27 (0.96–1.66) |
Cancer subtypes |
GGT (IU/L) |
|
|
|
|
|
|
|
Oral cavity |
M<63,W<35 |
1,436,534 (75.4) |
682 |
9,949,374 |
6.85 |
1 (Ref.) |
1 (Ref.) |
1 (Ref.) |
|
M≥63,W≥35 |
467,842 (24.6) |
267 |
3,216,695 |
8.30 |
1.48 (1.28–1.71) |
1.41 (1.22–1.63) |
1.44 (1.24–1.68) |
Pharynx |
M<63,W<35 |
1,436,534 (75.4) |
850 |
9,948,896 |
8.54 |
1 (Ref.) |
1 (Ref.) |
1 (Ref.) |
|
M≥63,W≥35 |
467,842 (24.6) |
302 |
3,216,676 |
9.39 |
1.39 (1.21–1.59) |
1.33 (1.16–1.53) |
1.34 (1.17–1.54) |
Larynx |
M<63,W<35 |
1,436,534 (75.4) |
793 |
9,948,717 |
7.97 |
1 (Ref.) |
1 (Ref.) |
1 (Ref.) |
|
M≥63,W≥35 |
467,842 (24.6) |
292 |
3,216,613 |
9.08 |
1.58 (1.37–1.81) |
1.47 (1.28–1.70) |
1.50 (1.30–1.73) |
Salivary gland |
M<63,W<35 |
1,436,534 (75.4) |
290 |
9,950,222 |
2.91 |
1 (Ref.) |
1 (Ref.) |
1 (Ref.) |
|
M≥63,W≥35 |
467,842 (24.6) |
67 |
3,217,168 |
2.08 |
0.78 (0.60–1.02) |
0.80 (0.61–1.06) |
0.83 (0.63–1.10) |
M, male; F, female; BMI, body mass index; WC, waist circumference; TG, triglyceride. GGT, gamma-glutamyl-transferase; IR, incidence rate (per 100,000 person-years); HR, hazard ratio; aHR, adjusted hazard ratio; CI, confidence interval.
1Adjusted for age, sex, and BMI.
2Adjusted for age, sex, BMI, smoking, alcohol consumption, regular physical activity, and income.
3Adjusted for age, sex, BMI, smoking, alcohol consumption, regular physical activity, income, glucose, insulin, oral hypoglycemic agent, and diabetes mellitus duration.
Statistically significant values are marked in bold.
Comment #4. In Table 1, how to understand the p-value, the author should list the details of how they calculate the results, including the groups and the methods.
Response: Thank you very much for the insightful comments. In Table1, the ANOVA was used to determine differences between the means of continuous variables and chi-square test was used to evaluate differences in the proportion of categorical variables. We added the sentence in the 2.5. statistical analysis in Materials and Methods section.
[Results] (page 4, paragraph 1, line 156)
The ANOVA was used to determine differences between the means of continuous variables and chi-square test was used to evaluate differences in the proportion of categorical variables.
Comment #5. In Table 3, how to understand there are fewer cases of cancer in Non-Obese (BMI<25) when the FLI is bigger, but there are more cases of cancer in Obese (BMI>25) when the FLI is bigger?
Response: We appreciate the reviewer’s thoughtful comments. Taking the oral cavity cancer as an example, 46 cases out of 66,523 non-obese subjects with FLI ≥60 and 202 cases out of 417,777 obese subjects with FLI ≥60 were found. Because the population size is different, it should be interpreted by incidence rate (IR) and adjusted hazard ratio (aHR).
In this case of oral cavity cancer, IR (per 100,000 person-years) was 10.12 and 6.97 for non-obese and obese with FLI ≥60, respectively. aHR was 1.46 and 1.22 for non-obese and obese with FLI ≥60, respectively. Therefore, the influence of NAFLD on HNC was more prominent in the non-obese patients than obese patients.
Table 3. The aHR and 95% CI for HNC in DM patients based on the combination of obesity and fatty liver.
Cancer subtypes |
Subtype |
FLI |
Number of subjects (%) |
Cases |
Duration (person-years) |
IR |
aHR (95% CI) |
p-value |
Oral cavity |
Non-Obese (BMI < 25) |
< 30 |
646,975 (34.0) |
292 |
4,427,638 |
6.60 |
1 (Ref.) |
0.040 |
< 60 |
269,541 (14.2) |
171 |
1,855,863 |
9.21 |
1.27 (1.05–1.54) |
|
||
≥ 60 |
66,523 (3.5) |
46 |
454,504 |
10.12 |
1.46 (1.06–2.00) |
|
||
Obese (BMI ≥ 25) |
< 30 |
128,038 (6.7) |
48 |
901,301 |
5.33 |
0.93 (0.68–1.26) |
|
|
< 60 |
375,522 (19.7) |
190 |
2,628,976 |
7.23 |
1.14 (0.95–1.37) |
|
||
≥ 60 |
417,777 (21.9) |
202 |
2,897,788 |
6.97 |
1.22 (1.01–1.47) |
|
||
Pharynx |
Non-Obese (BMI < 25) |
< 30 |
646,975 (34.0) |
376 |
4,427,475 |
8.49 |
1 (Ref.) |
0.009 |
< 60 |
269,541 (14.2) |
242 |
1,855,684 |
13.04 |
1.19 (1.01–1.41) |
|
||
≥ 60 |
66,523 (3.5) |
66 |
454,470 |
14.52 |
1.25 (0.96–1.63) |
|
||
Obese (BMI ≥ 25) |
< 30 |
128,038 (6.7) |
43 |
901,325 |
4.77 |
0.78 (0.57–1.08) |
|
|
< 60 |
375,522 (19.7) |
193 |
2,628,991 |
7.34 |
0.91 (0.76–1.08) |
|
||
≥ 60 |
417,777 (21.9) |
232 |
2,897,627 |
8.01 |
0.94 (0.80–1.12) |
  |
||
Larynx
|
Non-Obese (BMI < 25) |
< 30 |
646,975 (34.0) |
380 |
4,427,251 |
8.58 |
1 (Ref.) |
< 0.001 |
< 60 |
269,541 (14.2) |
233 |
1,855,635 |
12.56 |
1.08 (0.91–1.27) |
|
||
≥ 60 |
66,523 (3.5) |
75 |
454,440 |
16.50 |
1.32 (1.03–1.70) |
|
||
Obese (BMI ≥ 25) |
< 30 |
128,038 (6.7) |
37 |
901,307 |
4.11 |
0.80 (0.57–1.12) |
|
|
< 60 |
375,522 (19.7) |
145 |
2,629,063 |
5.52 |
0.72 (0.59–0.87) |
|
||
≥ 60 |
417,777 (21.9) |
215 |
2,897,633 |
7.42 |
0.91 (0.77–1.08) |
  |
||
Salivary gland |
Non-Obese (BMI < 25) |
< 30 |
646,975 (34.0) |
109 |
4,427,975 |
2.46 |
1 (Ref.) |
0.345 |
< 60 |
269,541 (14.2) |
62 |
1,856,110 |
3.34 |
1.27 (0.92–1.74) |
|
||
≥ 60 |
66,523 (3.5) |
9 |
454,605 |
1.98 |
0.80 (0.40–1.60) |
|
||
Obese (BMI ≥ 25) |
< 30 |
128,038 (6.7) |
29 |
901,348 |
3.22 |
1.45 (0.96–2.21) |
|
|
< 60 |
375,522 (19.7) |
70 |
2,629,258 |
2.66 |
1.10 (0.82–1.49) |
|
||
≥ 60 |
417,777 (21.9) |
78 |
2,898,094 |
2.69 |
1.20 (0.89–1.62) |
  |
IR, incidence rate (per 100,000 person-years); aHR, adjusted hazard ratio; CI, confidence interval; HNC, head and neck cancer; DM, diabetes mellitus; FLI, fatty liver index; BMI, body mass index. Adjusted for age, sex, BMI, smoking, alcohol consumption, regular physical activity, income, glucose, insulin, oral hypoglycemic agent, and DM duration.
Comment #6. In Table 3, please let us know the mean of N (%).
Response: We changed the term of ‘N (%)’ to ‘Number of subjects (%)’. Thank you for the comments.
Comment #7. In Table 3, please list the unit of “Duration”.
Response: We appreciate the reviewer’s comments. The unit of “Duration” is person-years. We modified Table 3 and added the unit of duration.
Comment #8. In figure 1, the quality is low, please change to a clearer one.
Response: We modified figure 1 to a 300 dpi, .tif image file. We appreciate the reviewer’s suggestion.

Round 2
Reviewer 1 Report
The Authors satisfactorily addressed all criticisms raised during the peer-review process. In this revised form, the manuscript is worthy of publication in your journal.
Reviewer 2 Report
Agree to accept.